# Analysis of the Temperature Distribution in a Refrigerated Truck Body Depending on the Box Loading Patterns

**DOI:** 10.3390/foods10112560

**Published:** 2021-10-23

**Authors:** Jun-Hwi So, Sung-Yong Joe, Seon-Ho Hwang, Soojin Jun, Seung-Hyun Lee

**Affiliations:** 1Department of Smart Agriculture Systems, Chungnam National University, Daejeon 34134, Korea; sjha24@cnu.ac.kr (J.-H.S.); hbs3689@naver.com (S.-H.H.); 2Department of Biosystems Machinery Engineering, Chungnam National University, Daejeon 34134, Korea; dhrmaksdyd@naver.com; 3Department of Human Nutrition, Food and Animal Sciences, University of Hawaii at Manoa, Honolulu, HI 96822, USA

**Keywords:** refrigerated truck, CFD modelling, temperature prediction, airflow, loading pattern

## Abstract

The main purpose of cold chain is to keep the temperature of products constant during transportation. The internal temperature of refrigerated truck body is mainly measured with a temperature sensor installed at the hottest point on the body. Hence, the measured temperature cannot represent the overall temperature values of transported products in the body. Moreover, the airflow pattern in the refrigerated body can vary depending on the arrangement of loaded logistics, resulting temperature differences between the transported products. In this study, the airflow and temperature change in the refrigerated body depending on the loading patterns of box were analyzed using experimental and numerical analysis methods. Ten different box loading patterns were applied to the body of 0.5 ton refrigerated truck. The temperatures inside boxes were measured depending on the loading patterns. CFD modeling with two different turbulence models (*k*-*ε* and SST *k*-*ω*) was developed using COMSOL Multiphysics for predicting the temperatures inside boxes loaded with different patterns, and the predicted data were compared to the experimental data. The *k*-*ε* turbulence model showed a higher temperature error than the SST *k*-*ω* model; however, the highest temperature point inside the boxes was almost accurately predicted. The developed model derived an approximate temperature distribution in the boxes loaded in the refrigerated body.

## 1. Introduction

The global cold-chain market has been rapidly grown because of the significant increase in fresh produce and vaccine logistics [1,2]. The key for cold chain is to maintain the temperature of the transported product at a constant low temperature [3]. A variety of refrigerated trucks are employed in cold-chain logistics. Refrigerated trucks are positioned as a key transportation system that connect all distribution processes [4]. A refrigerated truck is equipped with a refrigerated body and a refrigeration unit. The refrigerated body should have good thermal insulation performance to transport refrigerated cargo at low temperature [5]. In addition, refrigerated trucks should be equipped with a temperature monitoring system. Although different types of refrigeration systems are used depending on the type of refrigerated truck, a simple on–off feedback control with hysteresis is the most used system to control the temperature inside the refrigerated body depending on the setting temperature [6,7]. The temperature of the refrigerated body is measured only at the hottest location using a temperature sensor [8]. However, the measured temperature cannot represent the overall temperature information of the transported product by refrigerated truck [9]. A variety of cargoes are randomly loaded in refrigerated trucks. As the amount of cargo loaded in the refrigeration body increases, the space required for cold air circulation is reduced, finally preventing uniform cooling of randomly loaded cargo [10]. The on–off temperature control method causes temperature variation within the setting temperature, resulting in nonuniform temperature distribution in the refrigerated body [11]. The temperature variations are also affected by the location of the evaporator, the direction of supply and intake of cold air, and the arrangement of the cargo [12]. The quality of fresh produce can be deteriorated by this variation during transportation [13]. Therefore, monitoring the overall temperature distribution in the refrigerating body is necessary. By determining the temperature distribution in the body, fresh produce can be protected from being exposed to an inappropriate transportation environment [14].

A data logger or RFID tag equipped with a wireless temperature sensor is suitable for monitoring cargo because of its simple installation and small size [15,16]. A number of studies on wireless sensors have been conducted to prove the effectiveness and efficiency of the device by measuring the temperature change of cargo during transportation [17,18,19,20]. Real-time cargo monitoring using such sensors enables rapid response to temperature issues through accurate temperature data collection. If the temperature issue in the cargo is quickly resolved, the quality of the cargo can be maintained, and the amount of cargo discarded due to deterioration in quality can be reduced [21]. Therefore, wireless sensors are widely used for real-time cargo management in cold-chain transportation logistics [22,23]. However, since the real-time cargo monitoring system requires several supplemental devices such as temperature sensors and data loggers, it is more suitable for high value-added products (i.e., vaccines) than agricultural products with relatively low added value [24,25,26]. Therefore, it is necessary to develop a new temperature monitoring system that can measure the temperature value by the location of the cargo without using sensors in the cold-chain distribution [8].

A number of computational fluid dynamics (CFD) modeling methods have been developed to predict temperature distribution in the refrigerated trucks, and the number of sensors required for measuring temperature can be reduced based on the results from the modeling [27]. CFD modeling with experimental analysis methods have been conducted to analyze the cold airflow and temperature distribution in the refrigerated body of the truck. Yildiz, T. [12] performed airflow and temperature profile simulations according to cold air inflow and outflow models in a fully loaded trailer. The airflow circulation and the temperature difference between measurement points depending on cold air supply conditions were analyzed through the simulation. The results demonstrated that the proper place of the inlet and outlet of cold air could improve the temperature gradient in the refrigerated body. Kayansayan et al. [28] performed numerical analysis on turbulent flow with complex heat transfer in a refrigerated body to analyze the effect of shape factors of the body. By considering the two types of cold air outlets, the three types of aspect ratios, and the Reynolds number of the wind speed, it was possible to determine the optimum cold-air speed value depending on the shape of the refrigerated body. However, experimental work should accompany these methods to develop accurate modeling. Margeirsson et al. [29] developed a 3D heat transfer model that could be used to predict product temperature changes in supercooled cod fillets packaged in loaded EPS boxes. The developed model was validated by comparing with experimental results, and was slightly different from the experimental results. Hoang [30] presented CFD-based refrigerated vehicle heat transfer modeling to predict the temperature of cargo. Various phenomena such as respiration heat of the product and infiltration of outside air due to the opening of the door were considered in the CFD modeling. Through the developed model, it was technically feasible to quickly evaluate the effect of phenomena such as external temperature, wall insulation, and thermostat setting on the load.

In this study, the internal environment of the refrigerated body depending on box loading patterns was determined through the experiments and simulations. Based on the results from experiments, turbulent flow and heat transfer models were developed using a numerical analysis method. Two turbulent models (SST *k*-*ω* and *k*-*ε* models) were employed in the simulation for comparing the accuracy of the simulation and the calculation time. The locations with relatively high temperatures were predicted by analyzing the temperature distribution in the refrigerated body depending on box loading pattern.

## 2. Materials and Methods

### 2.1. Refrigerated Body Set Up

A refrigerated body for a 0.5-ton commercial refrigeration truck was used in this study as shown in Figure 1a. The refrigerated body was made of a sandwich insulation wall, which consisted of rigid polyurethane foam and aluminum panels, and the size of the body was 1950 mm in length, 1210 mm in width, and 1140 mm in height. The thickness of the front and top of the insulating wall and the other walls were 80 mm and 70 mm, respectively. A commercial refrigeration system (TSU015L, KD FROZEN COMPANY, Korea) consisting of an evaporator, compressor, condenser, and refrigeration controller was installed in the body. The size of the evaporator, consisting of a cold air supply fan and two intake fans, was 437 mm in length, 765 mm in width, and 210 mm in height. The cooling capacity of the refrigeration system was 705 kcal/h on 2 HP single phase. A controller was used to adjust the temperature in the refrigerated body and the defrost cycle. The volumetric flow rate of cold air was 30 m^3^/min. Cargo boxes used in cold-chain logistics are usually made of the various materials such as plastic, wood, and corrugated cardboard [31]. Commercial corrugated cardboard boxes were used for all experiments and were loaded into the body in different loading patterns. The dimension of the boxes were 350 mm × 300 mm × 300 mm (thickness of 3 mm). The empty boxes were used to determine airflow patterns depending on box loading patterns in the refrigerated body. Thermocouples were installed in the center of the empty boxes to measure the internal temperature.

### 2.2. Experimental Procedure

All experiments were started at 11 a.m. and the temperature values inside boxes were measured for 4 h. The set temperature and temperature drop (reoperating temperature) values of the refrigeration system were −15 °C and 1.5 °C, respectively. The defrost period was set to a 6 h interval. The external environment could not be controlled because the refrigerated body was placed outdoors. The experiment was conducted from May to June in 2020, and the average outside temperature from 11 a.m. to 1 p.m. was 20 °C (36.37° N 127.37° E 0 m, Daejeon, Korea).

Figure 1b shows the locations of where the temperature sensors were installed. The sensor for the temperature control of the refrigerated body was installed on the bottom of the intake fan of the evaporator. Two temperature and humidity probes (0572 6172, Testo, Germany) were installed on the outside of refrigerated body and at the center of front wall inside the refrigerated body to measure temperature and humidity every 5 s. In addition, all data points were monitored and recorded in a data logger (176 H2, Testo, Germany). Thermocouples (KK-K-30, OMEGA Engineering, USA) were installed at intervals of 50 mm throughout the refrigerated body to measure the supply air, intake air, and left, right, and rear air-temperature values. The temperature was measured and recorded every 5 s using a data logger (U6-Pro, LabJack, USA) connected to PC. Thermocouples were installed in the center of each box, and the temperature was also measured every 5 s. Figure 1c shows the connection configuration of the temperature and humidity sensors, data loggers and PC.

### 2.3. Box Loading Patterns

Figure 2 shows the 10 box loading patterns and experiments were performed using patterns (1) to (7) with 3 to 9 boxes. The loading patterns were loading methods that were not taken into account in actual transport situations. The experimental loading pattern was established to clearly reveal the temperature patterns depending on the positions of the boxes. Pattern (1) was employed to the develop numerical model. The experimental and simulated data obtained from patterns (2) to (7) were compared to verify the developed model. The temperature distributions in patterns (8) to (10) were predicted based on the validated model. When the boxes were loaded in the refrigerated body, they were arranged as densely as possible. However, there was still gaps between the boxes. The measured average gap was approximately 5 mm. In the simulation, the lateral spacing between the boxes was also set to 5 mm.

### 2.4. Numerical Analysis

In this study, numerical analysis was conducted to investigate the internal airflow and temperature in the refrigerated body. A computational fluid dynamics (CFD) model was developed to analyze cold airflow and temperature inside boxes loaded in the refrigerated body by using COMSOL Multiphysics (COMSOL 5.5, COMSOL, Inc., Palo Alto, CA, USA). In order to develop a practical CFD model in the refrigerated body loaded with the boxes, the modeling was simplified through the following assumptions.

The air inside the body was assumed to be a uniform and incompressible fluid. The air flow was considered as a constant turbulent flow because only the section in which the refrigeration system operates was considered.

Airflow and heat radiation from the outside of the body were neglected. Instead, the heat flow due to thermal convection from the outside of the body surface was considered, and a constant outside temperature was set on body surface.

Density, heat capacity, and thermal conductivity of the boxes’ and refrigeration body’s materials were all considered constant. Only air was considered as a function of temperature.

Heat from the refrigeration cycle was neglected. The cargo was designed as a closed box with an empty interior and no heat source.

Figure 3 shows the change in the supply air temperature of the refrigeration system used in the experiment. A temperature sensor connected to the temperature controller was installed on the intake fan. The set temperature in the refrigerated body was set to −15 °C, and the temperature drop (reoperating temperature) value of the refrigeration system was set to 1.5 °C to prevent overload of the refrigerator. In the refrigeration cycle, the refrigeration system turns on and off repeatedly depending on the set temperature. When the temperature of the intake fan fluctuated between −13 and −15 °C, the temperature of the outlet fluctuated between −10 and −17 °C. Only the ON cycle was considered to analyze the internal temperatures of the boxes when cold air was supplied to a maximum of −17 °C. The temperature data of the supply air obtained through the experiment were used for the simulation. A time-dependent study simulating patterns (1) to (7) was performed to determine the temperature changes in the boxes. The datapoints were collected from 0 to 220 s in 5 s intervals.

#### 2.4.1. Geometry

Figure 4a shows a schematic diagram of refrigerated body used in this study. Two geometries were designed: geometry 1, considering the thickness of the wall, and geometry 2, omitting wall. All dimensions of the refrigerated body, refrigerator, and cargo box were the same as the refrigerated body used in this study. The box domain was also filled with air. The domain of geometry 1 consisted of the airflow area inside the refrigerated body, the airflow area inside the box, and the solid area of the insulation wall. The wall domain was composed of the aluminum panel and rigid polyurethan foam. An evaporator was installed on the ceiling in front of the inside of the body in an asymmetrical position close to the left wall. The domains of geometry 2 excluded the domain of the insulating wall of geometry 1. As shown in Figure 4b, cold air was supplied to the rear direction of the refrigerated body and returned to the intake fan of the evaporator after absorbing the heat in the refrigerated body. The inner space and the fan of the evaporator were removed to simplify the model. Figure 4b also shows the temperature measurement points in the simulation. They were taken at the same locations as where the temperature was measured in the experiments. Figure 4c shows the created mesh domain for two geometries. The mesh size was decided in consideration of analysis time and model accuracy. The total number of elements was about 150,000. Free tetrahedral and triangular elements were used in mesh domain construction. The average element quality was 0.64. The input and output mesh domains of the evaporator were constructed with very fine mesh for increasing the accuracy of the model. The mesh of boxes adjacent to other domains, such as other boxes or walls of refrigerated bodies, was finer. The number of meshes was varied depending on the boundary domain and the number of boxes. Boxes have different meshes depending on the domain around the boundaries. A sensitivity analysis for the meshes was performed.

#### 2.4.2. Fluid Dynamics Model

The cold air rapidly evacuating from the evaporator at approximately 5 m/s created turbulent flow inside the refrigerated body. The Reynolds-Averaged Navier–Stokes (RANS) method was used to analyze the turbulent flow. The RANS equation was expressed as;
(1)$$\rho \frac{\partial u}{\partial t}+\rho \left(u\cdot \nabla \right)u=\nabla \cdot [-pI+\mu \left(\nabla u+{\left(\nabla u\right)}^{T}\right]$$
(2)ρ∇·u=0
where *ρ* is the density (kg/m^3^), *u* is the velocity (m/s), and *p* is the pressure (Pa).

Representative turbulence models were the *k*-*ε* model for calculating the center of a turbulent flow (high Reynolds number) and the *k*-*ω* model for calculating the flow close to the wall (low Reynolds number) [32]. Menter [33] proposed the shear stress transport (SST) *k*-*ω* model combining the advantages of the *k*-*ε* model and the *k*-*ω* model. For CFD modeling of refrigerated body, the SST *k*-*ω* model was found to be the most accurate model for predicting turbulent flow [34]. The *k*-*ε* model and the SST *k*-*ω* model were employed in this study, and the accuracy and analysis time of the developed models were compared. The *k*-*ε* turbulence model has two main dependent variables: turbulent kinetic energy *k* and turbulence dissipation rate *ε* [35].
(3)$$\mu_{T}=\rho C_{\mu }\frac{k^{2}}{\varepsilon }$$
where *μ_T_* is the turbulent viscosity (Pa·s) and *C_μ_* is turbulence model parameter.

The SST *k*-*ω* model equation introduced two dependent variables: turbulent kinetic energy, *k*, and turbulent specific dissipation rate, *ω* [33]. The turbulent viscosity in the SST *k*-*ω* was given by:(4)$$\mu_{T}=\frac{\rho a_{1}k}{\max\left(a_{1}\omega ,Sf_{v2}\right)}$$
where *S* is the characteristic magnitude of the mean velocity gradients, *f_v2_* is the interpolation function, and *a_1_* is the turbulent model constants. All parameters of the fluid dynamics model were selected in [33,35]

#### 2.4.3. Heat Transfer Model

The heat transfer equation for the air domains inside the refrigerated body and the boxes, and the solid domain of the insulation wall was expressed as;
(5)$$\rho C_{p}\frac{\partial T}{\partial t}+\rho C_{p}u\cdot \nabla T+\nabla \cdot \left(-k\nabla T\right)=0$$
where *ρ* is the density (kg/m^3^), *Cp* is the heat capacity (J/kg∙K), *k* is the thermal conductivity (W/m∙K), and *T* is the temperature (K).

#### 2.4.4. Boundary Conditions

The initial value of the internal air temperature was the temperature at which the refrigerator starts operating (i.e., the highest temperature in the refrigeration cycle). The air velocity at the inlet and outlet was considered constant. The outlet air velocity of the evaporator measured by an anemometer was 4.8 m/s. The turbulence variables at the inlet boundary were expressed as;
(6)$$k=\frac{3}{2}{\left(U_{ref}l_{T}\right)}^{2},\;\omega =\frac{k^{1/2}}{{\left(\beta_{0}{}^{*}\right)}^{1/4}},\;\varepsilon =C_{\mu }{}^{3/4}\frac{k^{3/2}}{L_{T}}\;U_{ref}=U_{0}=4.8m/s$$
where *l_T_* is the turbulence intensity and *L_T_* is the turbulent length scale.

The turbulence intensity and turbulent length scale applied in the simulation were 0.05 (5%) and 0.01, respectively. The intake fan was not implemented in the simulation and was set as a simple outlet. A fixed pressure of 101.3 kPa (atmospheric pressure) was assigned to the intake fan boundary. The temperature of cold air was same as the measured temperature value from the outlet of the evaporator. As shown in Figure 4a, the outer and inner boundaries of the insulation wall of geometry 1 were set with aluminum, and the gap between the boundaries was set with rigid polyurethane foam. The boundaries of the walls in geometry 2 were assigned to rigid polyurethane foam. A heat flux condition by convective heat transfer was applied to the external wall surface in contact with the external environment. The heat flux equation at the outer boundary due to convection was given by;
(7)$$-n\cdot \left(-k\nabla T\right)=h\left(T_{exp}-T\right)$$
where *h* is the convective heat transfer coefficient (W/m^2^∙K), *T_exp_* is the external temperature (K), and *n* is the normal vector on the boundary.

The average temperature value obtained from the experiments was applied to the external surface temperature. In addition, the convective heat transfer coefficient was set to 10 W/m^2^∙K for all walls based on the temperature data collected by measuring the temperature difference at the walls of the refrigerated body.

The cargo box domain (corrugated cardboard) was created as a geometry with no thickness. The thickness of the box layer (3 mm) was very thin compared to the length of the refrigerated body (1950 mm). Corrugated cardboard was surrounded by air. The box boundary was considered as the thin layer. Heat flux through the thin box boundary was given by;
(8)$$-n\cdot q=-\rho C_{p}\frac{\partial T}{\partial t}-\nabla \cdot q_{c}\;q_{c}=-k\nabla T$$
where *q* is the heat flux, ∇*T* is the temperature gradient, the heat capacity (*Cp*) of corrugated cardboard is 1260 J/kg∙K, density (*ρ*) is 20 kg/m^3^, and heat conductivity (*k*) is 0.07 W/m∙K

### 2.5. Error Calculation

The validity of the developed modeling was determined by comparing experimental and simulated data through the root mean square error (RMSE). RMSE (°C) was given by;
(9)$$RMSE=\sqrt{\frac{1}{n}\overset{n}{\underset{i=1}{\sum }}{\left(y_{p}{}^{i}-y_{m}{}^{i}\right)}^{2}}$$
where *y_p_* is the predicted temperature value (°C) by the simulation, *y_m_* is the measured temperature value (°C) from the experiments, and *n* is the number of values.

## 3. Results and Discussion

### 3.1. Experimental Results for Loading Pattern (1)

The temperature distribution obtained from the experiment for box loading pattern (1) is shown in Figure 5. Only this experiment was conducted in July 2020, and the outside average temperature was 35 °C. The temperature of the refrigerated body was decreased up to the setting temperature (−15 °C) of the intake fan of the evaporator. After about 220 s, the temperature of the intake fan reached −15 °C. Then, the refrigeration system stopped working and the internal temperature of the refrigerated body was sharply increased, owing to the heat transfer from the insulation wall and the refrigeration system. When the temperature increased by the setting temperature difference (−1.5 °C), the refrigeration system started to operate again. The temperature just before the refrigeration system stopped operating was the lowest, and there was a difference in temperature at the temperature measurement points in the body. The average operating cycle time of the refrigeration system, except for the first cooling period, was 220 s to reach the setting temperature. The lowest temperature values at the measurement points are listed in Table 1. Among temperature values, the measured temperature at the rear wall (point (5)) was the lowest; however, the temperature difference between this temperature and the cold air from the evaporator (point (1)) was about 2 °C. The temperature values of the side walls (point (3) and (4)) were slightly higher than the setting temperature. Nonuniform temperature distribution was observed in the refrigerated body. The temperature values in the boxes depending on the measurement points (point (6), (7), and (8)) were also different due to the temperature nonuniformity in the refrigerated body. The significant temperature difference was observed between the upper (point (6)) and lower (point (8)) box and was over 3 °C.

Figure 6 shows the temperature change in the refrigerated body and box while the refrigeration system was operating. There was a time delay between the temperature at the center of the box and the temperature inside the refrigerated body. The time delay was the time required for cold air to reach the center of the box, with an average delay of 60 s. As more boxes were loaded in the refrigerated body (loading patterns (2) to (7)), the operating time of the refrigeration system was decreased, and the average operating time was 110 s (data are not shown). Although the operating time was decreased, the time required for the box to reach the lowest temperature was longer than the operating time, with an average time of 220 s. As shown in Figure 3 and Figure 6b, the initial temperature values (refrigeration system was on) of the cold air (−10 °C) and the temperature in the box (−10 °C) were used for the initial input temperature values in the simulation. Since the supply air temperature for nine boxes did not vary significantly depending on the loading patterns, the temperature value of supply air was set as the function of time, and the final temperature was −17 °C as shown in Figure 6b. It ended at 120 s in the experiment but extended to 245 s in the simulation.

### 3.2. Numerical Analysis Results for Loading Pattern (1)

#### 3.2.1. Air Velocity Field

Numerical analysis was performed for three different cases depending to the types of geometries and turbulence models, with loading pattern (1): the geometry one and SST *k*-*ω* model (case one), geometry one and *k*-*ε* model (case two), and geometry two and *k*-*ε* model (case three). Figure 7 shows the air flow inside the refrigerated body for each case. In case one and case two, the velocity streamlines in the y-z plane were almost identical. When comparing the x-y planes, only the SST *k*-*ω* model showed a small eddy current at each edge, but the overall velocity distribution in the two different models was almost same. Even in the overall flow of the 3D streamline, there was little difference between the two turbulence models. The observed airflow in case two and case three was very similar, and the absence of the outer wall slightly affected the analysis of the airflow of the refrigerating body with the loading pattern (1).

Sensitivity analysis was performed for normal size and fine size as shown in Figure 8. The number of mesh elements in the normal-sized scenario was about 150,000, and when the element size was made fine, about 450,000 elements were created. The results from sensitivity analysis for normal- and fine-sized meshes are shown in Figure 9. As the mesh size became finer, the temperature values at the measured points were slightly different. However, there was not a significant difference between the results from the sensitivity analysis for fine- and normal-sized meshes. A very fine element was required when adjacent domains existed, but a normal size in the simulation was sufficient for predicting the overall airflow in the refrigerated body loaded with different box loading patterns.

#### 3.2.2. Temperature Field

The experimental result for loading pattern (1) was compared with the simulated results from case one, two, and three (Figure 10). In Figure 10, *E* and *S* denote experiment and simulation, respectively. The internal temperature, which started from −10 °C, was decreased over time. In the experiment, the lowest and highest temperature values were measured at the rear wall (point (5)) and right wall (point (3)), respectively. In all simulation cases, the simulated temperature values for a refrigeration operating time of 220 s were slightly different from the experimental temperature values; however, the minimum and maximum temperature values were consistent. Figure 10b shows the temperature change inside the box for each case. The initial value of the temperature inside the box was set to be the same as the experimental value. The temperature inside the box was slowly decreased because the inside of the box was not in direct contact with the cold air. Table 2 shows the simulated temperature values for all cases. The temperature values were different depending on the temperature measurement points. The highest difference between experimental and simulated temperature values was observed at point (5). The temperature difference for case one, two, and three were 1.46 °C, 1.5 °C, and 1.2 °C, respectively. The leading cause of the temperature difference was that the door was not implemented in the simulation. The door side of the refrigerated body has relatively weak insulation performance compared with the other walls. At all other points except for point five, the temperature difference between experimental and simulated results was less than 1 °C. Comparing cases two and three, the internal temperature range in case three was much larger, showing the relatively high temperature difference. The omitted wall in the simulation caused the temperature difference. The effect of heat transfer by convection was increased in the absence of insulating walls in the simulation geometry. Although the temperature values near the walls obtained in case three were different from the simulated results of cases one and two, the temperature change of the boxes in case three (Figure 10b) was not significantly different from cases one and two. The temperature values inside the boxes were not significantly affected by the presence of insulating walls in the simulation geometry. The predicted temperature values in the boxes were slightly deviated from the experimental data. This error was expected because the box geometry used in the simulation was different from the actual box geometry used in the experiment. The calculated RMSE values for the measured temperature values from different points are listed in Table 3. In all cases, some RMSE values were greater than 1 °C, but the majority of RMSE values were less than 1 °C. The modeling results could be considered reasonable. Table 4 shows the analysis times for the simulation of the refrigerated body with loading pattern (1) and pattern (2) in the three cases. The SST *k*-*ω* model was relatively more accurate than the *k*-*ε* model. However, the analysis time was significantly increased as the number of boxes was increased. Therefore, in terms of calculation time, the *k*-*ε* model would be practical for developing the prediction model for temperature distribution in the refrigerated body with different loading patterns. Case three, in which the insulation walls were omitted, predicted almost the same temperature values inside the boxes as case one and two, and shortened the calculation time.

### 3.3. Comparison between Experimental and Simulated Results for Loading Patterns (2) to (7)

Case three (geometry two, *k*-*ε* turbulence model) was used to predict the temperature change in the refrigerated body with loading patterns (2) to (7). Figure 11a shows the simulated temperature change in the refrigerated body with loading pattern (2). The rear side of the refrigerated body, which was in direct contact with cold air, cooled rapidly. The front side was not cooled enough because of the penetrated heat from the outside of the front wall. Compared to the ceiling and rear walls, cold air did not influence the side and front walls. Figure 11b shows the change in the temperatures inside the boxes over the simulation time. In the experiment, the initial temperatures of the boxes were different depending on the location. In the simulation, the initial temperature inside the box was set to −10 °C. The boxes on the top layer were the most affected by the cold air. However, since the boxes on the bottom layer were in contact with the bottom of the refrigerated body, the penetrated heat from the bottom affected the temperature of the bottom-layer boxes. The temperatures of middle-layer boxes were not significantly changed. Figure 11c shows the temperature change depending on the position of the box. In the experiment, it took about 50 s for the cold air to reach the center. The temperature difference depending on the box layers was observed. The lower-layer boxes showed a higher temperature than the initial value. Figure 12 shows the predicted temperature distribution in the refrigerated body with loading patterns (3) to (7). The temperature inside the boxes, which started at −10 °C, decreased over time. However, after 220 s, the core temperatures in the boxes did not reach the final cold air temperature (−17 °C). In loading patterns (2) to (7), the temperature values of bottom-layer boxes were much higher than in boxes in other layers. This was caused by the heat transfer from the bottom of the refrigerated body. The amounts of cooling and heat transfer from outside of the walls were different depending on the position of boxes. Because all boxes were loaded at the bottom of the refrigerated body in loading pattern (7), the amount heat transfer heat from the outside of bottom significantly affected the cooling inside the boxes and caused nonuniform temperature distribution. The predicted temperature distribution results were compared with the experimental temperature data measured in the center of the boxes. The average lowest temperature values of the boxes measured for 1 h are listed in Table 5. The temperature of cold air was decreased up to −17 °C in the experiment. However, the temperature values inside the boxes were not decreased below −12.5 °C. These temperature values clearly showed that the box loading patterns significantly affected temperature distribution in the refrigerated body. The standard deviation of all temperature values was below 0.5. Table 6 is the simulated temperature distribution in the refrigerated body depending on loading patterns. SImilarly to the experimental results, the simulated temperature in box seven of loading pattern (7) was the highest temperature and was increased up to −5.37 °C. Box 6 and Box 13 of loading pattern (4), which showed the largest temperature difference in the experiment, did not differ significantly by 1.4 °C in the simulation. Even though there was a difference between experimental and simulated temperature data, the simulated data were reasonable because the difference between the experimental value and the error were less than 1 °C in almost all simulations. Table 7 shows the ranking of the lowest temperature inside the boxes in experiments and simulations. The three boxes with the highest temperature in all patterns were identical in both experiments and simulations. In loading pattern (5), box 11 was predicted to be the lowest temperature box, but box 8 was the lowest. In loading pattern (6), box 8 was predicted to be the lowest temperature, but box 11 was the lowest in the experiment. However, since boxes 8 and 11 were adjacent boxes, these differences are reasonable errors and did not affect the overall temperature distribution in the simulation. In the loading patterns (2), (3), (4), and (7), the lowest temperature values in the boxes were almost same as the experimental data. The case three model, which had a relatively high temperature error near the wall, had a low box-temperature error. The developed model was able to predict the point at which the temperature would rise by analyzing the temperature distribution.

### 3.4. Prediction of Hot Spots during Cooling for Loading Patterns (8) to (10) Using the Developed Model

The developed model could not accurately predict the exact internal temperature of the boxes. Depending on the loading pattern, a location where the temperature of the box increased occurred, and the developed model was sufficient to identify hot spots in the refrigerated body. The developed model was employed to predict the highest temperature points for loading patterns (8) to (10) (Figure 13). In loading pattern (8), the front and middle boxes in the bottom layer were not cooled down enough. In the boxes of patterns (9) and (10), the temperatures of the boxes in the bottom layer were significantly increased as more boxes were loaded. In particular, the boxes near front side of the refrigerated body had a higher temperature than the boxes loaded at the rear side. The temperature distribution inside the refrigerated body loaded with 27 boxes was analyzed depending on the changes in the parameters of the refrigeration cycle. Figure 14 shows the application of the developed model. By changing the experimental conditions, which were difficult to set up in the experiment, it was possible to predict the box that was not cooled enough in the loading pattern. The supply-air flow rate and temperature did not affect the cooling of boxes loaded in the refrigerated body, although the outside temperature did. The temperature distribution in the refrigerated body filled with boxes was significantly affected by the outside temperature. The internal temperatures of the boxes predicted by the developed model was not as accurate as the temperature measured by the data logger. However, it was possible to predict the box location where cooling was not enough by the developed model.

## 4. Conclusions

Temperature monitoring systems using data loggers are widely used to manage cargo during transportation. However, as the amount of cargo increases, the more data loggers are required, causing the problems in cost and inconvenient installation. In this study, the model that could predict the temperature without a data logger was developed. Through the experiment, it was confirmed that the temperature was different depending on the location of the loaded boxes. Based on the experimental results, a temperature distribution model was developed for different loading patterns in the refrigerated body. For practical modeling, the *k*-*ε* model was selected as the turbulence model. In addition, the insulated walls and outside airflow were neglected in the modeling. In the developed model, the error increased when measurements were closer to the wall of the refrigerator body, but it was within an acceptable error range. The developed model in this study was able to rapidly predict the highest temperature points in the refrigerated body loaded with different box loading patterns by analyzing the temperature distribution of boxes loaded at random locations in various refrigeration systems. This developed model can be applied in the future, taking into account the loading patterns, cargo, and box sizes for real transport environments.

## Figures and Tables

**Figure 1 foods-10-02560-f001:**
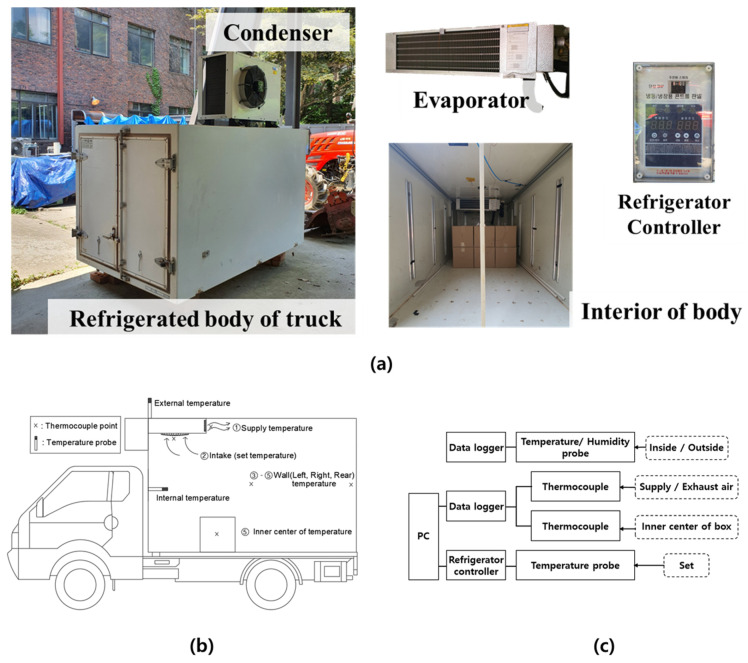
(**a**) Configuration and appearance of the refrigeration system (**b**) Installation position of temperature sensor (**c**) Configuration diagram of sensor and device.

**Figure 2 foods-10-02560-f002:**
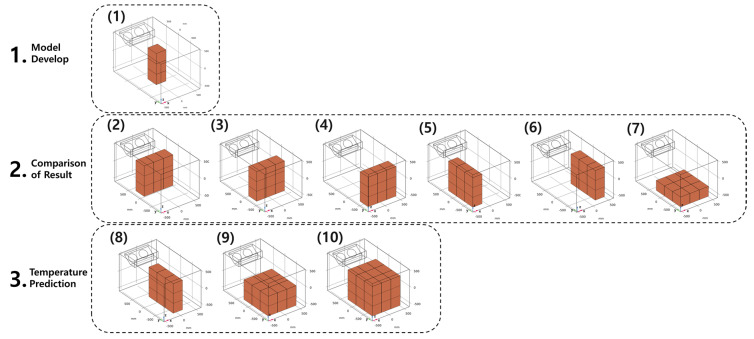
Box loading patterns used in experiments and numerical analysis.

**Figure 3 foods-10-02560-f003:**
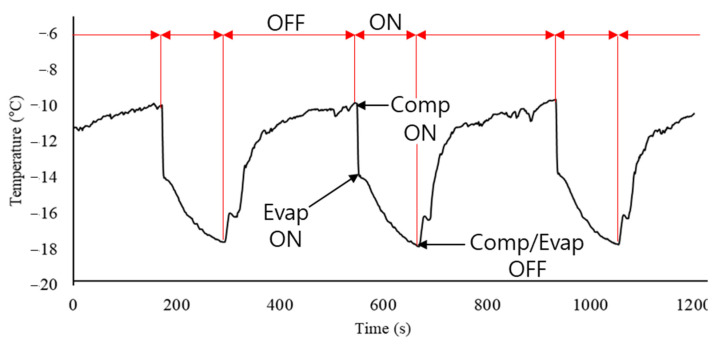
Change in supply air from the refrigeration system.

**Figure 4 foods-10-02560-f004:**
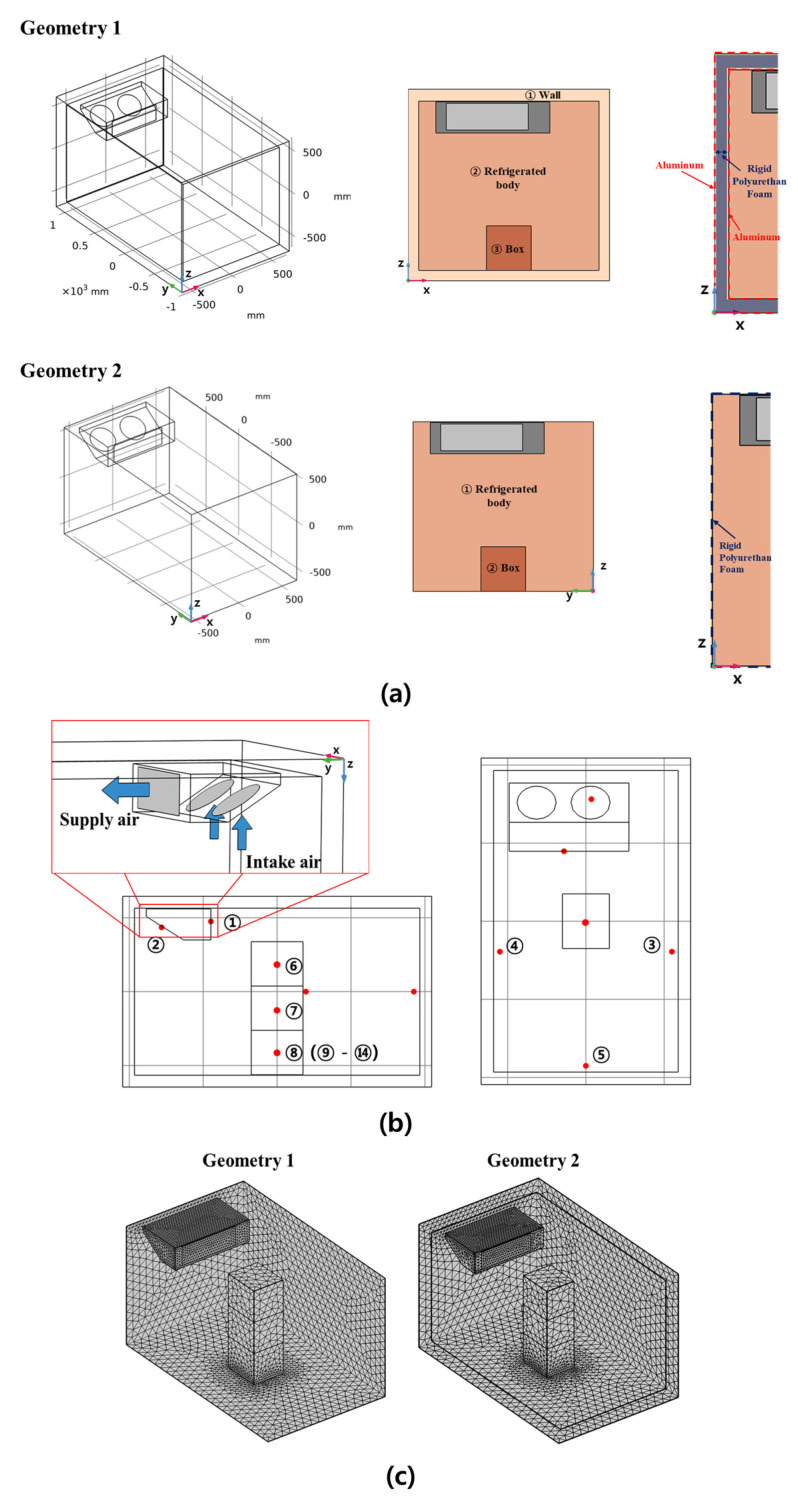
(**a**) Geometries of refrigerated body used in the simulation (1) Body with insulation wall (2) Body without insulation wall (**b**) Temperature measurement points and refrigerator boundary conditions in simulation (**c**) grid mesh of model.

**Figure 5 foods-10-02560-f005:**
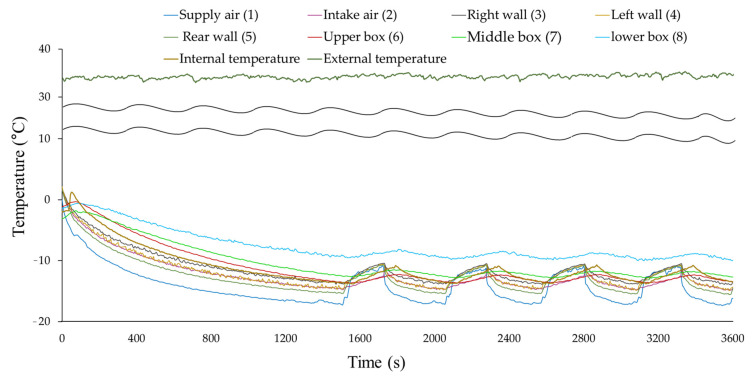
Temperature change in the refrigerated body with loading pattern (1): air (supply and intake fan), wall (5 cm from left, right, and rear wall), boxes (inner center point of upper, middle, and lower boxes).

**Figure 6 foods-10-02560-f006:**
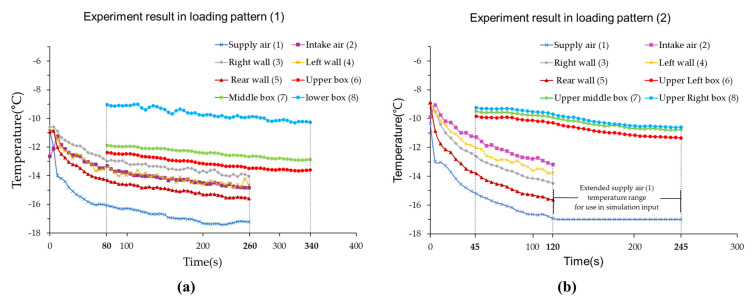
Temperature change in the refrigerated body when refrigeration systems were operating: (**a**) Temperature change by point in the operating cycle in the experiment with loading pattern (1), (**b**) Temperature of the supply air used in simulation with loading patterns (2).

**Figure 7 foods-10-02560-f007:**
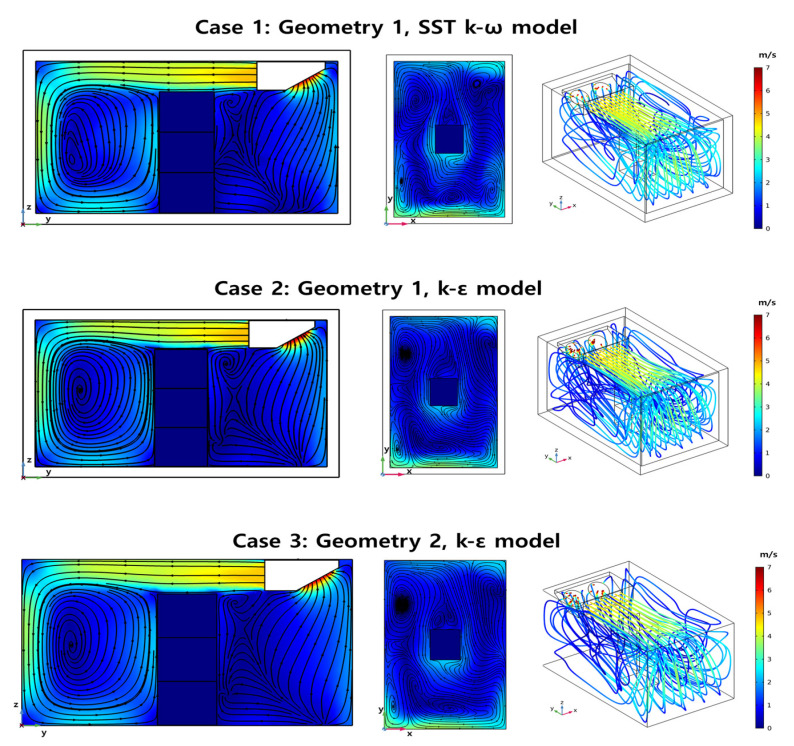
Numerical Analysis of Airflow depending on the cases (y-z plane: x = 0, x-y plane: z = 0).

**Figure 8 foods-10-02560-f008:**
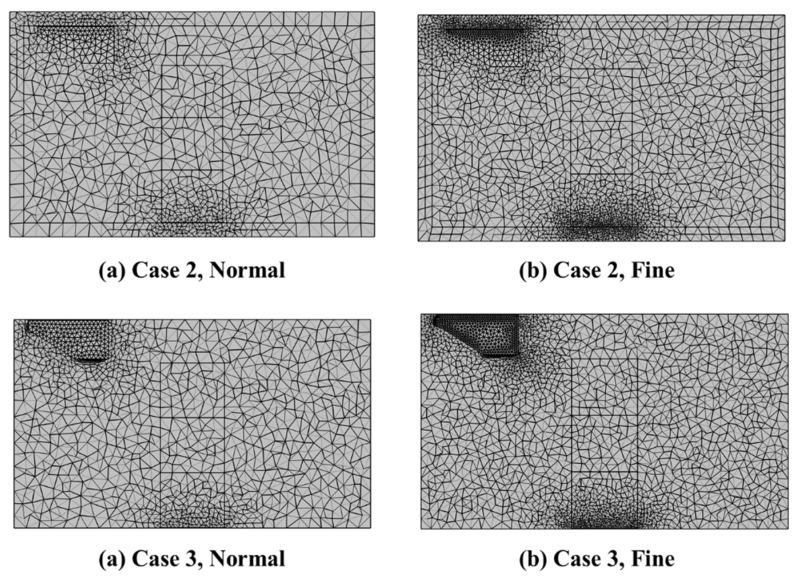
Two-element sized mesh grid for case 2 and case 3.

**Figure 9 foods-10-02560-f009:**
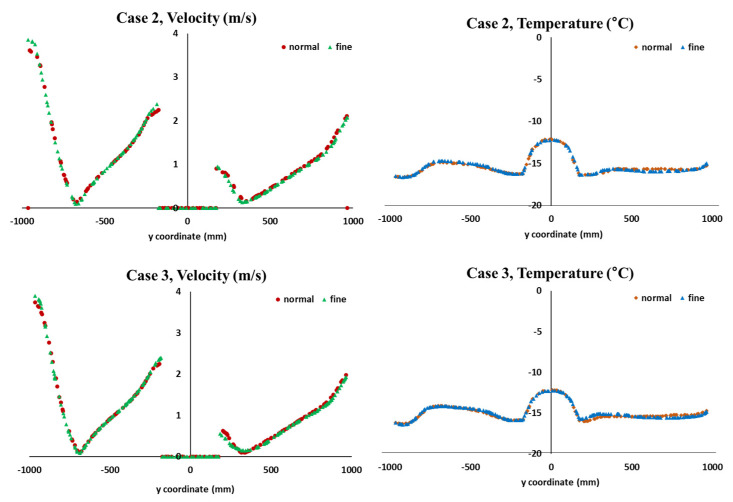
Comparison of velocity and temperature distribution depending on mesh size in y-coordinate (x = 0, z = 0).

**Figure 10 foods-10-02560-f010:**
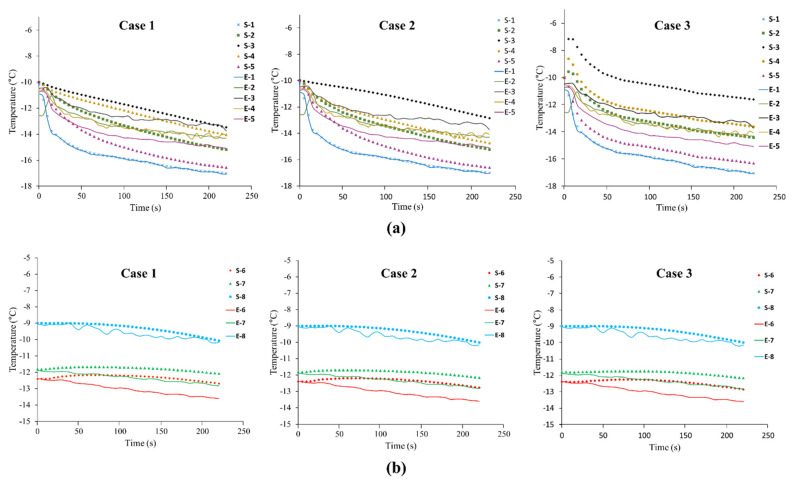
Comparison of experimental and simulated results (case 1, 2, and 3) for loading pattern (1): (**a**) temperature values in refrigerated body and (**b**) three boxes.

**Figure 11 foods-10-02560-f011:**
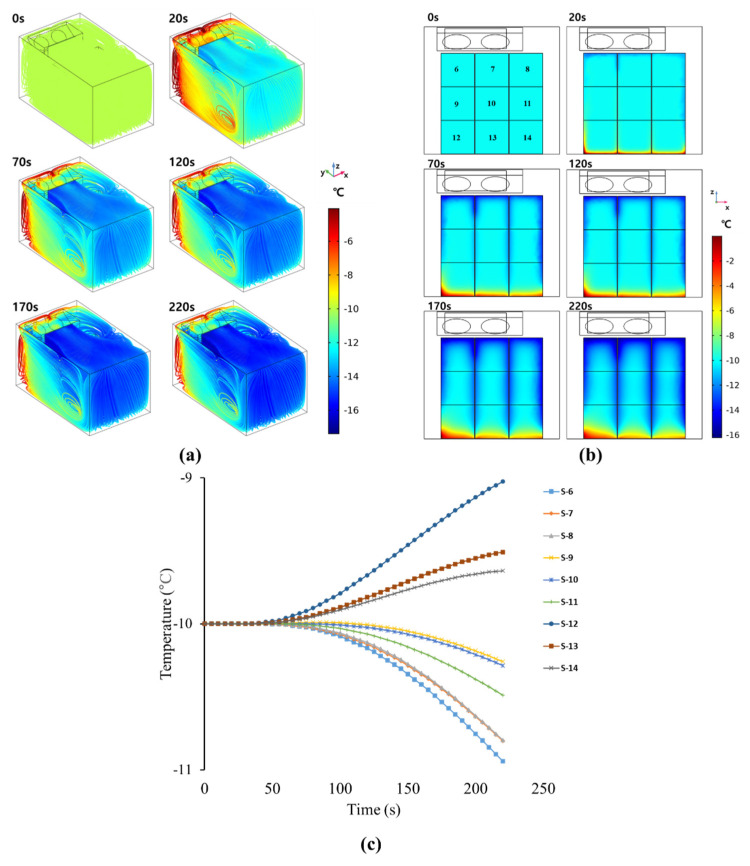
Temperature change over time: (**a**) 3D streamlines in the refrigerated body, (**b**) middle slide inside the boxes, and (**c**) inner center of the box at each box point (**b**) with time.

**Figure 12 foods-10-02560-f012:**
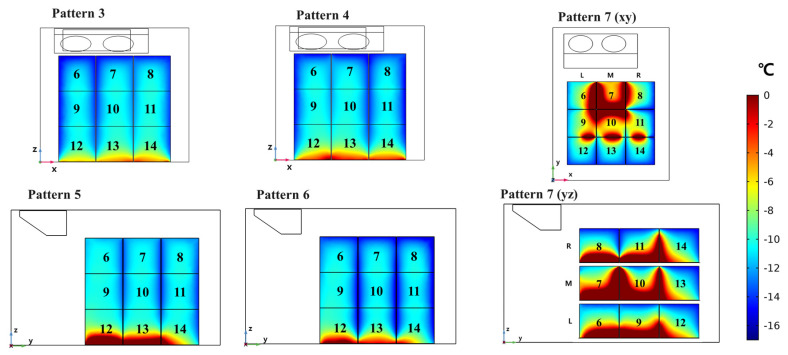
Predicted temperature distribution in boxes with different loading patterns.

**Figure 13 foods-10-02560-f013:**
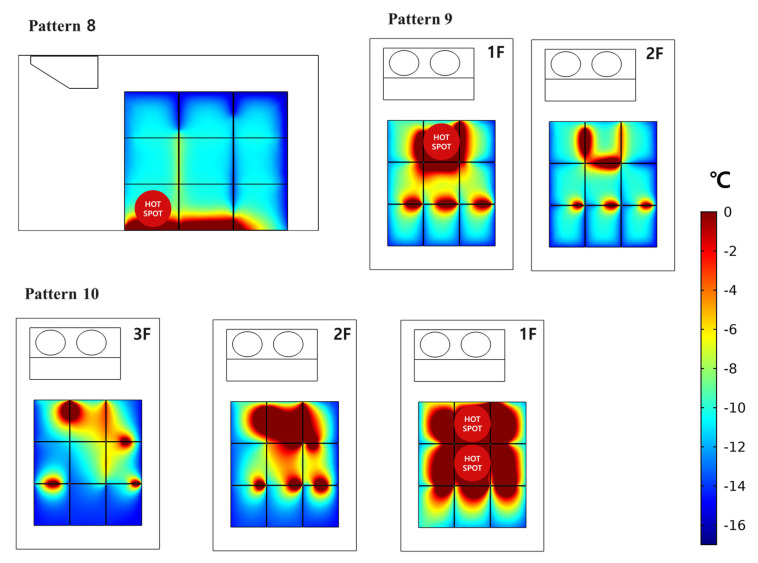
Predicted location of the box with the highest temperature depending on loading patterns (8) to (10).

**Figure 14 foods-10-02560-f014:**
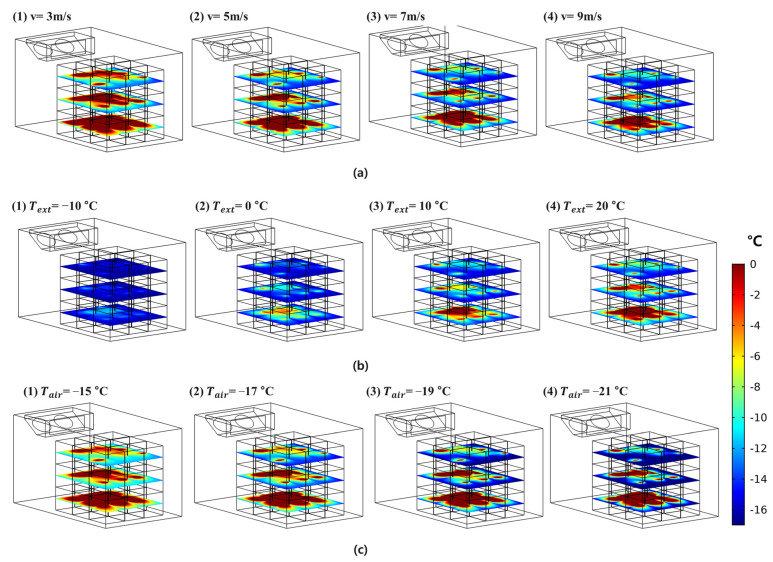
Application of the developed model; evaluation of the influence of experimental conditions; (**a**) change in supply-air velocity (*v*), (**b**) change in external temperature (*T_ext_*), and (**c**) change in cold air temperature (*T_air_*).

**Table 1 foods-10-02560-t001:** The lower temperature limits of measurement points (setting temperature: −15 °C).

Point	1	2	3	4	5	6	7	8
Temperature (°C)	−17.21	−14.83	−14.03	−14.63	−15.59	−13.59	−12.87	−10.26

**Table 2 foods-10-02560-t002:** Temperature values according to the point inside the refrigerated body and boxes at 220 s.

Simulation		Point							
	Case	1	2	3	4	5	6	7	8
Temperature (°C)	1	−17.00	−15.18	−13.47	−14.03	−16.55	−12.69	−12.08	−10.07
	2	−17.00	−15.24	−12.85	−14.74	−16.60	−12.77	−12.15	−10.01
	3	−17.05	−14.42	−11.62	−13.61	−16.30	−12.86	−12.16	−9.99

**Table 3 foods-10-02560-t003:** RSME values (°C) of predicted temperatures by the developed model.

Case	Point						
	2	3	4	5	6	7	8
1	0.724	0.754	1.082	0.980	0.758	0.570	0.303
2	0.718	1.242	0.515	0.998	0.714	0.531	0.316
3	0.700	2.173	1.030	1.027	0.653	0.512	0.334

**Table 4 foods-10-02560-t004:** Calculation times for the simulation of loading pattern (1) and pattern (2) in three cases.

Case	Loading Pattern	
	1	2
1	13 min 41 s	1 h 10 min 17 s
2	9 min 42 s	22 min 35 s
3	9 min 3 s	21 min 24 s

**Table 5 foods-10-02560-t005:** The average lowest temperature values of boxes temperature measured over 1 h.

Experiment	Pattern	Point								
		6	7	8	9	10	11	12	13	14
Average Temperature (°C) ± Stdev	2	−11.56	−11.36	−11.13	−10.69	−10.37	−10.89	−8.26	−8.34	−9.25
		±0.31	±0.44	±0.39	±0.37	±0.49	±0.38	±0.29	±0.27	±0.21
	3	−12.17	−11.99	−11.63	−11.57	−11.11	−11.55	−10.06	−9.87	−10.71
		±0.2	±0.29	±0.28	±0.28	±0.39	±0.3	±0.22	±0.27	±0.28
	4	−12.52	−12.31	−11.95	−11.72	−11.18	−11.50	−9.05	−8.66	−10.00
		±0.13	±0.2	±0.2	±0.19	±0.29	±0.27	±0.19	±0.23	±0.24
	5	−10.45	−10.69	−11.31	−10.25	−10.43	−11.01	−7.66	−8.28	−8.96
		±0.06	±0.04	±0.05	±0.08	±0.05	±0.07	±0.13	±0.07	±0.1
	6	−9.81	−10.16	−10.69	−10.11	−10.20	−10.69	−8.70	−8.98	−9.63
		±0.45	±0.5	±0.38	±0.42	±0.44	±0.4	±0.19	±0.24	±0.2
	7	−7.38	−6.61	−7.59	−7.18	−5.99	−7.76	−9.00	−8.96	−9.79
		±0.02	±0.05	±0.03	±0.03	±0.03	±0.01	±0.06	±0.07	±0.03

**Table 6 foods-10-02560-t006:** The simulated lowest temperature values in the boxes.

Simulation	Pattern	Point								
		6	7	8	9	10	11	12	13	14
Predicted Temperature (°C) ± Error *	2	−10.94	−10.80	−10.80	−10.26	−10.29	−10.49	−9.03	−9.51	−9.64
		0.62	0.56	0.33	0.43	0.09	0.41	−0.77	−1.17	−0.39
	3	−11.00	−10.75	−10.54	−10.68	−10.42	−10.18	−9.98	−9.40	−9.38
		1.17	1.24	1.09	0.89	0.69	1.37	0.08	0.48	1.33
	4	−10.77	−10.74	−10.85	−10.44	−10.37	−10.65	−9.47	−9.30	−9.61
		1.76	1.57	1.10	1.28	0.80	0.85	−0.42	−0.64	0.39
	5	−10.62	−10.57	−10.64	−10.35	−10.65	−10.65	−9.27	−9.62	−10.19
		−0.18	0.12	0.67	−0.10	−0.22	0.36	−1.61	−1.33	−1.23
	6	−10.77	−10.74	−10.85	−10.44	−10.37	−10.65	−9.47	−9.30	−9.61
		−0.96	−0.58	−0.16	−0.32	−0.17	0.04	−0.77	−0.31	0.02
	7	−7.27	−5.37	−7.73	−7.77	−6.40	−8.03	−9.58	−9.52	−9.69
		0.11	1.24	−0.14	−0.59	−0.41	−0.27	−0.59	−0.56	0.10

* Error = Predicted temperature—Measured temperature.

**Table 7 foods-10-02560-t007:** Ranking for the lowest center temperature in the boxes depending on loading patterns.

	Pattern	Method	Point								
			6	7	8	9	10	11	12	13	14
Temperature Rank *	2	Experiment	1	2	3	5	6	4	9	8	7
		Simulation	1	2	3	6	5	4	9	8	7
	3	Experiment	1	2	3	4	6	5	8	9	7
		Simulation	1	2	4	3	5	6	7	8	9
	4	Experiment	1	2	3	4	6	5	8	9	7
		Simulation	2	3	1	5	6	4	8	9	7
	5	Experiment	4	3	1	6	5	2	9	8	7
		Simulation	4	5	3	6	2	1	9	8	7
	6	Experiment	6	4	2	5	3	1	9	8	7
		Simulation	2	3	1	5	6	4	8	9	7
	7	Experiment	6	8	5	7	9	4	2	3	1
		Simulation	7	9	6	5	8	4	2	3	1

* In descending order of temperature.

## Data Availability

Data presented in this study are available in the article.

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
