# Peer review of "Analysis of the Temperature Distribution in a Refrigerated Truck Body Depending on the Box Loading Patterns"

_foods, 2021, doi:10.3390/foods10112560_

Round 1

Reviewer 1 Report

General comments

The paper presents CFD simulations of airflow and temperature profiles inside a small truck, loaded with cardboard boxes. It is not clear why the different patterns are considered and if they have any relevance to actual transport situations. The same criticism holds for the large temperature fluctuations that are seen. More importantly, it seems that essential elements (gap modelling and grid dependence) of the models are not described and explained so it is impossible to judge how some of the results (in particular the irregular temperature patterns in Figure 11 are a mystery) are obtained.

Specific comments

Line 120. Which material is inside the boxes? It is important to describe and define the thermo-physical properties that should be representative of frozen food products.

Line 155. It should be made clear why these patterns are considered. What is the relevance of this stackings for practical loads. I can imagine that no truck driver wants to drive around with load nr. 1-6- as they will soon collapse in the first turn or stop.

Line 176. The off cycle seems very relevant for the dynamics of this application, as it will allow to see if the prediction works well without the cooler on (which is the majority of the time). It is should be explained why it is not considered.

Line 179. This is a very large temperature fluctuation by the cooling system. Is this realistic for trucks?

Line 212. It is not needed to describe the complete turbulence model, which can be found in standard reference works. What is missing, is how the boundary layers are modelled. These are crucial to determine the heat transfer rates with the walls and boxes and will determine the dynamics. In addition it lust be demonstrated using a grid refinement study that the surface transfer rates are sufficiently accurately predicted. This is also particularly relevant for the gaps between the boxes: from plots like Fig 9 it seems that air flows between the boxes -otherwise such temperature profiles cannot be obtained. Please elaborate properly.

Line 334. Why is humidity plotted? It is not modelled so please remove. The numbers in the legend need to be explained in the figure caption. The reader should go look for another figure to understand this figure.

Line 340. The legend and caption of this figure needs to clarified. Now the figure is impossible to understand. Plots a and b seem to be switched.

Line 360. Geo 1 and 2 will not be different for airflow as the wall treatment does not affect the resulting airflow. If you see differences, they are numerical errors.

Line 502. It is a but a mystery how the irregular profiles Figure 11 can be obtained. There are real spots of high temperature on some interfaces between boxes, but not on others. It seems a random pattern and so in in principle not predictable with CFD, unless it would concern numerical errrors. It should be very carefully explained to the reviewers and in the text how these patterns arise and how valid these are.

Author Response

General comments

The paper presents CFD simulations of airflow and temperature profiles inside a small truck, loaded with cardboard boxes. It is not clear why the different patterns are considered and if they have any relevance to actual transport situations. The same criticism holds for the large temperature fluctuations that are seen. More importantly, it seems that essential elements (gap modelling and grid dependence) of the models are not described and explained so it is impossible to judge how some of the results (in particular the irregular temperature patterns in Figure 11 are a mystery) are obtained.

Answer: The authors appreciate the reviewer’s constructive comments and suggestions. The manuscript was revised to incorporate the review comments. As suggested by the reviewer, the relevant sentences have been rewritten and have been presented in different color (blue) in the revised manuscript. The responses to general and specific comments were as follows.

Specific comments

Line 120. Which material is inside the boxes? It is important to describe and define the thermo-physical properties that should be representative of frozen food products.

Answer: The boxes used in this study were empty. An empty box was employed to determine air flow patterns as function of box and body dimensions. Therefore, this study was focused to analyze the cooling and heating effect depending on the location of the boxes. For future study, the boxes will be filled with frozen food products. As suggested by the reviewer, the description of the inside of the box was added in the revised manuscript. (Line #144, #229)

Line 155. It should be made clear why these patterns are considered. What is the relevance of this stackings for practical loads. I can imagine that no truck driver wants to drive around with load nr. 1-6- as they will soon collapse in the first turn or stop.

Answer: In general, transporters want the cargo to be full. In addition, the patterns mentioned in the paper are not considered stable in most actual transportation situations. In the study, in order to analyze the effect of the location of the box on the temperature, the temperature was measured by simply stacking 9 boxes and changing the location. As suggested by the reviewer, the explanation about loading pattern was added. (Line #175)

Line 176. The off cycle seems very relevant for the dynamics of this application, as it will allow to see if the prediction works well without the cooler on (which is the majority of the time). It is should be explained why it is not considered.

Answer: This study focused on determining of the internal temperatures of the boxes when the set temperature was reached by cooling the interior with the refrigeration system. Therefore, in order to simplify the modeling, off-cycle was not considered in the simulation. As suggested by the reviewer, the explanation of why off-cycles were not considered was added in the revised manuscript. (Line #210)

Line 179. This is a very large temperature fluctuation by the cooling system. Is this realistic for trucks?

Answer: This temperature was the temperature measured in front of the outlet of the refrigerator. The refrigerator could set the temperature difference to be turned on/off. Since the off cycle was not considered in this study, the temperature difference of 1.5°C was set up to prevent overload of the refrigerator. The temperature sensor connected to the temperature controller was installed on the suction fan side of the refrigerator. The temperature drop (re-operating temperature) value of the refrigeration system was set to 1.5°C, but the temperature at the outlet was larger (Fig.5). As suggested by the reviewer, the explanation about temperature fluctuation was added in the revised manuscript (Line #210).

Line 212. It is not needed to describe the complete turbulence model, which can be found in standard reference works. What is missing, is how the boundary layers are modelled. These are crucial to determine the heat transfer rates with the walls and boxes and will determine the dynamics. In addition it lust be demonstrated using a grid refinement study that the surface transfer rates are sufficiently accurately predicted. This is also particularly relevant for the gaps between the boxes: from plots like Fig 9 it seems that air flows between the boxes -otherwise such temperature profiles cannot be obtained. Please elaborate properly.

Answer: As suggested by the reviewer, the equations of fluid simulations were removed with a few exceptions (Line #270). In addition, the equations and mesh used in the boundary layer were added in the revised manuscript (Line #240, #316).

Line 334. Why is humidity plotted? It is not modelled so please remove. The numbers in the legend need to be explained in the figure caption. The reader should go look for another figure to understand this figure.

Answer: Answer: As suggested by the reviewer, the plot and caption of Figure 5 was reformatted (Line #376).

Line 340. The legend and caption of this figure needs to clarified. Now the figure is impossible to understand. Plots a and b seem to be switched.

Answer: As suggested by the reviewer, the Fig. 6 was reformatted and the supplementary explanation was added (Line #370, #384).

Line 360. Geo 1 and 2 will not be different for airflow as the wall treatment does not affect the resulting airflow. If you see differences, they are numerical errors.

Answer: The speed distribution between case 2 and case 3 actually showed some difference. This was the reason why the steam line was different. However, the speed difference was within 0.2 m/s, which was judged to be the effect of the temperature error on the outer wall analyzed in the temperature part. In addition, the expression “same” was modified to “very similar” in the revised manuscript (Line #398).

Line 502. It is a but a mystery how the irregular profiles Figure 11 can be obtained. There are real spots of high temperature on some interfaces between boxes, but not on others. It seems a random pattern and so in in principle not predictable with CFD, unless it would concern numerical errrors. It should be very carefully explained to the reviewers and in the text how these patterns arise and how valid these are.

Answer: The high temperature was caused by heat coming in from the outside. The high temperature that appears in some areas was generated in a section that was not cooled by cold air. In fact, it was very complicated to experimentally validate predictions about the temperature gradient of the box. However, there was no significant error when only the core temperature was compared in the previous pattern verification step. In addition, as mentioned in Line #514, the position of the box with the highest central temperature per pattern could be predicted. Fig. 11 visually shows the temperature distribution for patterns 8-10 similar to the actual loading pattern. In addition, the description of Fig. 11 was added in the revised manuscript. (Line #538, #557)

Reviewer 2 Report

The authors pursue to replace a high number of sensors for monitoring goods in a refrigerated truck. Instead, temperatures should be predicted by CFD simulation and single/few sensors for e.g. supply air.

The paper is far from giving an experimental justification for this approach.

- The authors tested only empty boxes, which react in seconds or minutes to temperature changes. A realistic scenario with filled boxes, taking hours or days to cool down was not tested.

- Experimental results were only presented for pattern 1-7. Boxes had direct contact to the airflow at 2 or 5 sides. If there are more boxes in the truck, they are shielded on all sides by their neighbours, like pattern 10. Experimental results are missing here. Simulation is also inaccurate due to 10fold gap diameter.

- We observed high spatial temperature variation in trucks and containers in our tests, with hotspots at varying locations. Caused by unknown factors such as deviations of loading temperatures or varying gap diameters by careless stowage or box deformations. A simulation would always result in the same temperature distribution.

- The accuracy of the simulation is not sufficient according to figure 8b. The change of the experimental temperature over time is about 1°C. The error between simulation and measurement is about the same magnitude.

*** Section 2.2 experimental procedure

How many temperature and humidity probes?

*** Section 2.4.2 Fluid dynamics

There is no need to give all standard equations of fluid simulations. Better give only a reference to a textbook. Give only equations that you developed or modified yourself.

Line 231: Are these the standard setttings of Comsol for the parameters? If you modified them, on which basis?

Line 273: the turbulence intensity is a critical parameter for some models. How do you determine it? Did you carry out a sensitivity analysis? (How changes in the turbulence intensity affect the result)

Mesh-size / resolution / number of cells is also a missing critical parameter. Check also sensitivity to mesh size (if you double the number of cells, does is affect the result?)

*** Section 3.4 loading patters 8-10

Why did you change the gap diameter from 5mm to 50 mm. I guess it is about required mesh size.

Simulation with 10fold gap diameter does not give any useful results for temperature and cooling in the gaps.

Simulations were not compared with experimental results for the last 3 loading patters. I expect high deviations.

*** Further remarks

Please define ‚body‘. For me it is not a common word in refrigerated transportation.

You are considering partly loaded trucks. Please state in the introduction

Line 95: respiration heat is better than ‘breathing’

Line 125: the all experiment?

Author Response

The authors pursue to replace a high number of sensors for monitoring goods in a refrigerated truck. Instead, temperatures should be predicted by CFD simulation and single/few sensors for e.g. supply air.

Answer: The authors appreciate the reviewer’s constructive comments and suggestions. The manuscript was revised to incorporate the review comments. As suggested by the reviewer, the relevant sentences have been rewritten and have been presented in different color (blue) in the revised manuscript.

The paper is far from giving an experimental justification for this approach.

- The authors tested only empty boxes, which react in seconds or minutes to temperature changes. A realistic scenario with filled boxes, taking hours or days to cool down was not tested.

Answer: It was investigated how the location of the box affects cold and heat by excluding conditions such as the type of box and the food inside. Modeling was conducted to find the worst-case scenario in the distribution in which the box was actually loaded. In addition, A description of the box condition was added to the revised manuscript (Line #144).

- Experimental results were only presented for pattern 1-7. Boxes had direct contact to the airflow at 2 or 5 sides. If there are more boxes in the truck, they are shielded on all sides by their neighbours, like pattern 10. Experimental results are missing here. Simulation is also inaccurate due to 10fold gap diameter.

Answer: The developed model was verified through experimental patterns 1 to 7. Predictions for the remaining patterns 8 to 10 were performed using the validated model. Errors in temperature values might be occurred; however, it was shown that the model could be used to determine the degree of cooling depending on the location of the box. The reason the experiment was not performed is that at least 27 thermocouples are required for pattern 10. In addition, 10 fold gap figure was removed because it was far from our study purpose (Line #175).

- We observed high spatial temperature variation in trucks and containers in our tests, with hotspots at varying locations. Caused by unknown factors such as deviations of loading temperatures or varying gap diameters by careless stowage or box deformations. A simulation would always result in the same temperature distribution.

Answer: This study focused on identifying areas where crates were not properly cooled when boxes were loaded in the refrigerated body. The issues that may arise in the actual shipping environment mentioned by the reviewers were not considered in this study. Based on the results of this study, it will be considered in future studies. In addition, a description of the box loading pattern was added in the revised manuscript (Line #175).

- The accuracy of the simulation is not sufficient according to figure 8b. The change of the experimental temperature over time is about 1°C. The error between simulation and measurement is about the same magnitude.

Answer: We focused on the temperature difference between the boxes, not on an accurate temperature prediction. The error of the temperature prediction value was about 1 °C, but the temperature difference between the boxes was almost similar to the experimental value, so it was judged as a reasonable error. In addition, a description of the temperature error was added in the revised manuscript (Line #433).

*** Section 2.2 experimental procedure

How many temperature and humidity probes?

Answer: Two Temperature and humidity probe (one inside and one outside) was installed. In addition, the number of probes was added in the revised manuscript (Line #158).

*** Section 2.4.2 Fluid dynamics

There is no need to give all standard equations of fluid simulations. Better give only a reference to a textbook. Give only equations that you developed or modified yourself.

Answer: As suggested by the reviewer, the equations of fluid simulations was removed with a few exceptions (Line #270).

Line 231: Are these the standard setttings of Comsol for the parameters? If you modified them, on which basis?

Answer: These parameters are provided by Comsol and referenced from ‘Wilcox, D.C. Turbulence Modelling for CFD; DCW Industries: La Canada, CA, 1998;’

Line 273: the turbulence intensity is a critical parameter for some models. How do you determine it? Did you carry out a sensitivity analysis? (How changes in the turbulence intensity affect the result)

Answer: 0.05 (5%) is an ideal value for the turbulence intensity provided by Comsol. In ‘Alptekin, Ersin & Ezan, Mehmet & Kayansayan, Nuri. (2014). In Flow and Heat Transfer Characteristics of an Empty Refrigerated Container.’, the value of the turbulence intensity was 0.1 (10%). When the velocity distributions for the two values were compared, the two values did not differ significantly from each other.

Mesh-size / resolution / number of cells is also a missing critical parameter. Check also sensitivity to mesh size (if you double the number of cells, does is affect the result?)

Answer: As suggested by the reviewer, description and figure about created mesh in the model was added in the revised manuscript (Line #240). The finer the mesh, the longer the study time, so the size is similar in accuracy and shortens the study time.

*** Section 3.4 loading patters 8-10

Why did you change the gap diameter from 5mm to 50 mm. I guess it is about required mesh size.

Simulation with 10fold gap diameter does not give any useful results for temperature and cooling in the gaps.

Simulations were not compared with experimental results for the last 3 loading patters. I expect high deviations.

Answer: Constant stacking of boxes in the experiment produced some gaps, and the measured gap was 5mm. When the model developed in this study predicted the box temperature distribution for pattern 10, the boxes loaded with 5 mm were not cooled effectively. Therefore, 50 mm gap was applied to analyze the cooling effect by changing the box layout in the developed model. The last three loading patterns also analyzed the location where the cargo was not cooled by predicting the temperature distribution for the pattern with the developed model, and it was to show the use of the developed model. In addition, the pattern for 50mm was deleted because it was far from the objectives of this study (Line #184, #538)

*** Further remarks

Please define ‚body‘. For me it is not a common word in refrigerated transportation.

You are considering partly loaded trucks. Please state in the introduction

Answer: The reason we used the term “refrigerated truck body or refrigerated body” is because we only considered the loading pattern, not the actual transportation environment. As suggested by the reviewer, the explanation for “refrigerated body” was added in the revised paper (Line #53).

Line 95: respiration heat is better than ‘breathing’

Answer: As suggested by the reviewer, “breathing” was changed to “respiration” (Line #113).

Line 125: the all experiment?

Answer: As suggested by the reviewer, the sentence was corrected. (Line #141)

Reviewer 3 Report

In this study, the airflow and temperature change of the refrigerated body due to the box loading patterns were analyzed using experimental and numerical analysis methods. The study is very useful as it is an important step towards refrigeration system temperature prediction for cooling systems application.
The paper is well written and organized and contains enough data to support valid conclusions. Only minor issues should be addressed by the authors. These are as follows:

Materials and Methods

Ln 120-122
Please indicate that empty boxes were used. This is evident from the conclusions, but must be stated in the M&M section.

Figures

Fig. 5
Please explain the meaning of 2-8 in the figure description. I assume that 2-8 represent the locations for temperature measurement, temperature but it is not clear from the description.

Fig. 6
Please explain E1-E8 in the legend.

Fig. 9
Please explain the meaning of S-6 - S14 in the description of the figure.

Author Response

In this study, the airflow and temperature change of the refrigerated body due to the box loading patterns were analyzed using experimental and numerical analysis methods. The study is very useful as it is an important step towards refrigeration system temperature prediction for cooling systems application.

The paper is well written and organized and contains enough data to support valid conclusions. Only minor issues should be addressed by the authors. These are as follows:

Answer: The authors appreciate the reviewer’s constructive comments and suggestions. The manuscript was revised to incorporate the review comments. As suggested by the reviewer, the relevant sentences have been rewritten and have been presented in different color (blue) in the revised manuscript.

Materials and Methods

Ln 120-122

Please indicate that empty boxes were used. This is evident from the conclusions, but must be stated in the M&M section.

Answer: As suggested by the reviewer, the description of the inside of the box was added in the revised manuscript (Line #141, #229).

Figures

Fig. 5

Please explain the meaning of 2-8 in the figure description. I assume that 2-8 represent the locations for temperature measurement, temperature but it is not clear from the description.

Answer: As suggested by the reviewer, the legends were revised in the Figure 5 (Line #375).

Fig. 6

Please explain E1-E8 in the legend.

Answer: As suggested by the reviewer, the legends were revised in the Fig 6 (Line #383).

Fig. 9

Please explain the meaning of S-6 - S14 in the description of the figure.

Answer: As suggested by the reviewer, the caption about the Fig (c) was revised in the Fig 9 (Line #523).

Reviewer 4 Report

The objective of the study was the investigation of the temperature variations in a refrigerated truck body during transportation of food products, as a function of loading condition. The study is interesting and of high scientific value, since the stage of food transportation is a critical stage of the actual cold chain, where temperature exhibits significant deviations from the recommended range. The study is well designed and the data well interpreted from the engineering point of view. I recommend that the authors give more emphasis on the food aspect, indicating the interactions of the temperature fluctuations with food quality and safety markers. This may be achieved, for example, by applying the thermal transfer mathematical models on representative bacterial growth curves, applicable to perishable food products, or another food quality indicator high sensitivity in temperature.   

Author Response

The objective of the study was the investigation of the temperature variations in a refrigerated truck body during transportation of food products, as a function of loading condition. The study is interesting and of high scientific value, since the stage of food transportation is a critical stage of the actual cold chain, where temperature exhibits significant deviations from the recommended range. The study is well designed and the data well interpreted from the engineering point of view. I recommend that the authors give more emphasis on the food aspect, indicating the interactions of the temperature fluctuations with food quality and safety markers. This may be achieved, for example, by applying the thermal transfer mathematical models on representative bacterial growth curves, applicable to perishable food products, or another food quality indicator high sensitivity in temperature.

Answer: The authors appreciate the reviewer’s constructive comments and suggestions. Based on this study, we plan to conduct research by adding refrigerated cargo in the future. As suggested by the reviewer, we will add food quality and safety labeling to our study plan in the future.

Round 2

Reviewer 2 Report

The paper was improved in some minor points. However, the basic problems have not been handled.

The authors misunderstood the idea of a sensitivity analysis regarding the number of cells in the simulation. Goal is not to find the setting, which provides the fastest simulation time. Instead, a threshold should be found, for which a further refinement of the mesh does not affect the simulation results. For example, if the ‘fine’ and ‘extra fine’ mesh settings result in almost identical temperature and airflow patterns, it can be concludes that ‘fine’ is sufficient.

The authors used the ‘coarse’ mesh, which should only be used for first tests, but not for accurate simulation.

***

The authors write “The initial value of the temperature inside the box was set to be the same as the experimental value.” (line 371).

This in an acceptable approach. But therefore, the accuracy of the simulation can only be evaluated based on temperature changes, not on the absolute temperature, or temperature differences between boxes.

I have to insist on this point from my first review:

The change of temperature in the experimental data is about or less 1°C (Fig. 8b case 1). The RMSE between simulation and experiment is about the same magnitude (0.76, 0.57, 0.3 °C, Table 3) for the temperature in the boxes. At the end of the cooling period, the differences are even larger.

A simulation that fails to predict heat transfer and cooling effect is useless for analysing refrigerated systems.

The authors should have stopped here and refined the simulation.

***

In scenarios 8 to 10, the inner boxes are cooled by airflow through narrow gaps or heat transfer through neighbouring boxes.

The simulation has to be validated separately for this case. The case of scenario 1 were cooling mostly takes place by direct contact of the boxes with free airflow is physically different and cannot be taken for verification here. Especially if the first scenario already lacks accuracy as shown above.

I also have to insist on this point.  

***

In their first version, they wrote that they introduced extra wide gaps of 50mm in the simulation for scenarios 8 to 10. The problem behind it, might have been that narrow gaps need a very fine mesh.

Now they write that they reduced the gaps in the simulation to 5 mm. But the simulation results in Figure 12 do not show any difference. A reduction of gap width from 50 mm to 5 mm should largely affect the airflow between boxes and cooling.

It seems that the authors simply deleted information from the text without redoing the simulation.

***

Despite the inaccuracies, the simulation might help to identify boxes with a high probability for the lowest / highest temperature. But it is not possible to “predict the temperature without a data logger” as written in the conclusion.

***

Further remarks:

- the line numbers in the “Response to reviewer” are wrong.

- line 155 “to (10) were”

- line 243 please renumber equations

- line 343 should be figure 6

- line 393 Unit for RMSE is missing, also in Table 3

Author Response

The paper was improved in some minor points. However, the basic problems have not been handled.

The authors misunderstood the idea of a sensitivity analysis regarding the number of cells in the simulation. Goal is not to find the setting, which provides the fastest simulation time. Instead, a threshold should be found, for which a further refinement of the mesh does not affect the simulation results. For example, if the ‘fine’ and ‘extra fine’ mesh settings result in almost identical temperature and airflow patterns, it can be concludes that ‘fine’ is sufficient.

The authors used the ‘coarse’ mesh, which should only be used for first tests, but not for accurate simulation.

Answer: As suggested by the reviewer, the description of the mesh and the sensitivity analysis results were supplemented in the revised manuscript. (Line #209, #361)

***

The authors write “The initial value of the temperature inside the box was set to be the same as the experimental value.” (line 371).

This in an acceptable approach. But therefore, the accuracy of the simulation can only be evaluated based on temperature changes, not on the absolute temperature, or temperature differences between boxes.

I have to insist on this point from my first review:

The change of temperature in the experimental data is about or less 1°C (Fig. 8b case 1). The RMSE between simulation and experiment is about the same magnitude (0.76, 0.57, 0.3 °C, Table 3) for the temperature in the boxes. At the end of the cooling period, the differences are even larger.

A simulation that fails to predict heat transfer and cooling effect is useless for analysing refrigerated systems.

The authors should have stopped here and refined the simulation.

Answer: Clearly, the predictions for the change in the center temperature of the box in loading pattern (1) were not accurate. However, it was an acceptable error in consideration of the error that may occur in the experimental process and the error due to the simplification of the simulation. As the reviewer’s comment, the model did not predict the exact internal temperatures of boxes loaded in the refrigerated body, but it was enough to quickly identify the hot spots (where cooling was not enough) in 9 boxes. Based on this study, our future research goal is to develop the applications that can detect worst scenarios for larger systems and more cargo, as well as the 0.5 ton system used in the experiments. Even though the developed model had slightly less accuracy, the model was possible to quickly predict the hot spot in the refrigerated body. Therefore, as the reviewer suggested, the expression 'accurate temperature prediction' has been modified to clarify the accuracy and the aim of the model (Line #24, #499, #518). In addition, as suggested by the reviewer, an explanation for the error was added in the revised manuscript (Line #404).

***

In scenarios 8 to 10, the inner boxes are cooled by airflow through narrow gaps or heat transfer through neighbouring boxes.

The simulation has to be validated separately for this case. The case of scenario 1 were cooling mostly takes place by direct contact of the boxes with free airflow is physically different and cannot be taken for verification here. Especially if the first scenario already lacks accuracy as shown above.

I also have to insist on this point.

Answer: Although there is no verification process for the simulation, it is possible to predict the high temperature point based on the results obtained previously. However, since this model could not predict the exact temperature, the expression for 'temperature prediction' was changed to 'prediction of temperature rise point' (Line #520, #512, #521, #526).

***

In their first version, they wrote that they introduced extra wide gaps of 50mm in the simulation for scenarios 8 to 10. The problem behind it, might have been that narrow gaps need a very fine mesh.

Now they write that they reduced the gaps in the simulation to 5 mm. But the simulation results in Figure 12 do not show any difference. A reduction of gap width from 50 mm to 5 mm should largely affect the airflow between boxes and cooling.

It seems that the authors simply deleted information from the text without redoing the simulation.

Answer: The reason we initially changed the default spacing of 5mm to 50mm was not to consider the mesh size, but we would like to check the worst-case transport environment (sparsely loaded boxes). Similarly, Figure.12 also tried to check the temperature distribution of the boxes used as box loading pattern in real transport environment by the developed model.

By default, when a non-air region (such as another box) is adjacent to a box region, the mesh near the box boundary has a finer mesh than the air domain. The reason why 50mm was deleted is because it is a loading pattern not mentioned in the study and because other readers may be misunderstood as the reviewers thought. In addition, we recognized that 50mm would create a bigger problem with accuracy as a different mesh size than the default loading patterns would be created.

***

Despite the inaccuracies, the simulation might help to identify boxes with a high probability for the lowest / highest temperature. But it is not possible to “predict the temperature without a data logger” as written in the conclusion.

Answer: That was our research goal (Determining the highest temperature point in the refrigerated body rather than the exact temperature). As suggested by the reviewer, the expression for predicting the correct temperature in conclusion was revised in the manuscript (Line #538).

***

Further remarks:

- the line numbers in the “Response to reviewer” are wrong.

Answer: As suggested by the reviewer, the line numbers was checked.

- line 155 “to (10) were”

Answer: As suggested by the reviewer, the word was corrected (Line #155).

- line 243 please renumber equations

Answer: As suggested by the reviewer, Incorrect numbering was corrected (Line #244, #251, #260, #273, #285, #292).

- line 343 should be figure 6

Answer: As suggested by the reviewer, Incorrect numbering was corrected (Line #343).

- line 393 Unit for RMSE is missing, also in Table 3

Answer: As suggested by the reviewer, the unit of RMSE was added in the revised manuscript (Line #291, #409, Table 3).

Round 3

Reviewer 2 Report

Better check box temperature for scenario 8-10 in future reserach.